# Sensitive and Stable Electrochemical Sensor for Folic Acid Determination Using a ZIF-67/AgNWs Nanocomposite

**DOI:** 10.3390/bios12060382

**Published:** 2022-05-31

**Authors:** Yujiao Sun, Xue Wang, Hao Zhang

**Affiliations:** 1Beijing Laboratory of Food Quality and Safety, College of Food Science and Nutritional Engineering, China Agricultural University, Beijing 100083, China; b20203060458@cau.edu.cn; 2Department of Nutrition and Health, China Agricultural University, Beijing 100091, China; xwang326@cau.edu.cn

**Keywords:** folic acid, metal–organic framework, electrochemical sensor, stability

## Abstract

An electrochemical sensor using silver nanowires (AgNWs)-doped with a zeolite-like metal–organic framework (ZIF-67) was developed for highly sensitive and stable determination of folic acid (FA). The ZIF-67/AgNWs nanocomposite was prepared by a one-step reaction via a template method and drop-coated onto the surface of a screen-printed carbon electrode (SPCE) to form a ZIF-67/AgNWs@SPCE electrochemical sensing platform. The electrochemical square wave voltammetry (SWV) curve for this sensing platform was measured in an electrolyte solution containing FA under the optimum experimental conditions. The redox peak current of FA (*I*_FA_) increased with increases in the FA concentration (*C*_FA_). There was a linear relationship between *I*_FA_ and *C*_FA_ in the range of 0.1 μM to 10 μM, and the determination limit was 30 nM. The ZIF-67/AgNWs@SPCE was used as an electrochemical sensor for FA which maintained a good stability over 7 days and showed good determination performance in real samples with a high recovery rate (100.9–102.1%, *n* = 6).

## 1. Introduction

Metal–organic frameworks (MOFs) are crystalline porous materials with a periodic network structure formed by self-assembly of transition metal ions with organic ligands. These materials are of interest because of their high porosities, high specific surface areas, controllable morphologies, and chemical tunability [1]. Unfortunately, most pristine MOFs have low electrical conductivity, which limits their use in electrochemical sensing. Zeolite imidazole frameworks (ZIFs) are MOFs composed of transition metal cations and organo-imidazole ester junctions [2]. ZIFs have large specific surface areas and good chemical and thermal stability [3]. ZIF-67 consists of a metal ion (Co^2+^) and an organic compound (2-methylimidazole). Interestingly, ZIF-67 combines the benefits of both MOFs and conventional zeolites [4]. Therefore, the excellent stability makes it possible to synthesize ZIF-67 in an environmentally friendly and economical way. Without high temperature or high pressure, room temperature is enough for the reaction to synthesize well-formed framework structures. ZIF-67 and its derivatives have been studied for the usage in molecular separation [5], gas adsorption [6], electrochemical catalysis [7], and many other applications. Similar to conventional MOFs, ZIF-67 has insulating properties. Therefore, increasing its conductivity would be beneficial for its application in electrochemical sensing. Many different multifunctional MOF-based composites have been investigated to enhance the electrical conductivity. The performance of composites is usually superior to that of individual components because of the synergistic effect between functional units [1]. Combining other components or structures with ZIF-67 to make compounds with better properties than pure ZIF-67 is an effective approach, and it is of interest to explore the characteristics and applications of ZIFs [8].

One-dimensional metallic nanomaterials have been widely investigated for their unique electrical properties. As a result, these types of nanomaterials are widely used in optoelectronics and nanoelectronics because of their high aspect ratios and excellent densities [9,10]. Silver, in particular, has high electrical conductivity, and silver nanowires (AgNWs) show great potential for a wide range of practical applications [11]. Nanostructured silver often has poor stability and easily aggregates, which obviously limits its practical application. As conductive fillers for the preparation of AgNW conductive polymer nanocomposites, the usage of AgNWs retains the advantages of high surface area and high electrical conductivity, while compensates for the disadvantages of AgNWs [12,13].

Folic acid (FA), commonly known as vitamin B9, is a water-soluble vitamin [14]. Vitamins are organic substances found in foods and they are essential in small quantities for normal body functioning [15]. Serum FA levels below the 5 ng mL^−1^ indicate a deficiency, which leads to a lot of diseases [16]. FA in the blood is mainly present as 5-methyltetrahydrofolate, and a deficiency of it leads to hyperhomocysteinemia [17,18]. Lack of FA can also lead to a range of health problems in pregnant women [19]. The requirement for FA monitoring has driven development of a variety of analytical devices for FA determination using HPLC [20], spectrophotometric [21], colorimetric [22], and fluorometric [23] methods. However, these devices have some major disadvantages, such as high setup costs, time-consuming procedures, and the requirement for specialized knowledge for instrument operation [24].

Electrochemical sensing devices generally consist of two main components: the sensing material and the transducer. The sensing material interacts with the target and undergoes a chemical change (electron transfer), and the signal generated by this change is converted into a readable signal by a converter [25]. Compared to other optical sensing methods, such as the colorimetric method based on Lambert–Beer law, electrochemical sensors are simple to operate and have a higher sensitivity. Recently, nanomaterials have been applied to the design of electrochemical sensors to improve the analytical performance and the stability of electrochemical sensors [26]. For example, Venu et al. synthesized Co_3_O_4_/reduced graphene oxide (rGO) nanocomposites for the construction of electrochemical sensors for the determination of FA, which exhibited effective electron-mediated behavior with a determination limit of 19 μM. However, the material was synthesized under high pressure and high temperature [27]. Abdelwahab et al. synthesized gold nanoclusters (AuNCs)/activated graphene (AGR)/MWCNT nanocomposite, which could be obtained by electrochemical reduction with an enhanced current response to FA oxidation with a determination limit as low as 0.09 μM. Additionally, the electrochemical sensor was used for the determination of real samples [28]. Electrochemical sensors have been developed for FA monitoring [29,30,31,32,33,34], but these methods always have complex and costly preparation conditions, and the stability of the prepared sensors need to be improved.

Herein, we developed an electrochemical sensor using a ZIF-67/AgNWs nanocomposite for stable and sensitive determination of FA (Figure 1). The ZIF-67/AgNWs nanocomposite was prepared by a one-step reaction via a template method and drop-coated onto a screen-printed carbon electrode (SPCE) to form a ZIF-67/AgNWs@SPCE electrochemical sensing platform. FA underwent an oxidation reaction, which increased the redox peak current of FA (*I*_FA_). We explored the use of the ZIF-67/AgNWs@SPCE sensor to detect FA by plotting the linear relationship between *I*_FA_ and the FA concentration (*C*_FA_). The sensor showed stable and sensitive determination performance in biological samples.

## 2. Materials and Methods

### 2.1. Chemicals

AgNWs were purchased from Leader-Nano (Jining, China, http://www.leadernano.com/ (accessed on 26 November 2021)). Uric acid (UA), 2-Methylimidazole, Co(NO_3_)_2_·6H_2_O, and FA were purchased from Shanghai Maclean Biochemical Technology Co. (Shanghai, China, http://www.macklin.cn/ (accessed on 29 November 2021)). Glucose and anhydrous methanol were purchased from Tianjin Yongda Reagent Co. (Tianjin, China, http://www.tjydhxsj.com/ (accessed on 20 October 2021)). Ascorbic acid (AA) was purchased from Beijing Aoboxing Biotechnology Co. (Beijing, China, https://bjabx.company.lookchem.cn/ (accessed on 15 August 2021)). Analytical grade NaCl, KCl, and CaCl_2_ were purchased from Shanghai Sinopharm Chemical Reagent Co. (Shanghai, China, https://www.sangon.com/ (accessed on 20 October 2021)). Serum samples, vitamin B_1_, vitamin B_2_ were purchased from Beijing Solarbio Science & Technology Co. (Beijing, China, http://solarbio.bioon.com.cn/ (accessed on 17 January 2022)). All chemicals and reagents were used directly without purification. Double distilled water and phosphate-buffered saline containing 0.9% NaCl (PBS) were used for pH adjustment in the experiments.

### 2.2. Apparatus

Scanning electron microscopy (SEM) images and Energy dispersive spectrometer (EDS) were obtained using a ZEISS Gemini SEM 300 (Jena, Germany, https://www.zeiss.com.cn/microscopy/home.html (accessed on 9 January 2022)). Fourier transform infrared (FT-IR) spectra were measured using a PerkinElmer Frontier (Akron, OH, USA, https://www.perkinelmer.com.cn/ (accessed on 14 December 2021)). UV-vis absorption spectra were measured using a UV-2550 spectrophotometer (Shimadzu, Kyoto, Japan, https://www.shimadzu.com.cn/ (accessed on 17 January 2022)). Powder X-ray diffraction (XRD) was performed using graphitized monochromatic Co Kα radiation by a Rigaku Ultima IV (Rigaku, Tokyo, Japan, https://www.nihonkohden.com/ (accessed on 31 December 2021)). X-ray photoelectron spectroscopy (XPS) was performed using a K-Alpha spectrometer (Thermo Fisher Scientific, Waltham, MA, USA, https://www.thermofisher.cn/cn/zh/home.html (accessed on 2 January 2022)). Raman spectra were measured using a LabRAM HR Evolution micro confocal Raman spectrometer (HORIBA Jobin Yvon S.A.S., Palaiseau, France, https://www.horiba.com/int/ (accessed on 26 December 2021)). Electrochemical profiles in cyclic voltammetry (CV) and square wave voltammetry (SWV) modes were detected using a CHI-1080C electrochemical workstation (Shanghai, China, http://www.chinstr.com/ (accessed on 8 December 2021)). Electrochemical impedance spectra (EIS) were measured using a CHI-760E electrochemical workstation (Shanghai, China). A three-electrode system using a modified SPCE was used for electrochemical measurements. Each electrochemical was repeated six times and the results were averaged.

### 2.3. Preparation of the ZIF-67/AgNWs Nanocomposites

ZIF-67/AgNWs were synthesized in one step using a template method. First, 1.5 mL 5 mg mL^−1^ AgNWs was slowly added dropwise to a methanol solution containing 109.3 mg dimethylimidazole under stirring for 20 min to obtain mixture A. Next, 87 mg Co(NO_3_)_2_·6H_2_O was dissolved in 10 mL methanol to obtain solution B. Afterward, solution B was added dropwise to solution A and the resulting mixture was stirred vigorously for 1 min. After stirring, the mixture was left to stand for 24 h at room temperature [35,36]. After the reaction was finished, the mixture was centrifuged at 12,000 rpm for 15 min. The resultant precipitate was repeatedly washed with methanol and then dried under vacuum at 40 °C. The purified ZIF-67/AgNWs complex was dispersed in methanol (1 mg mL^−1^) for subsequent use. Blank ZIF-67 was prepared using the same method but without AgNWs.

### 2.4. Preparation of ZIF-67/AgNWs@SPCE

The solution of ZIF-67/AgNWs (5 μL) was dropped on the surface of a SPCE, which was then dried in air at room temperature, the sensing platform based on ZIF-67/AgNWs@SPCE was achieved.

### 2.5. Electrochemical Measurements and Electrochemical Sensor Fabrication

CV, EIS, and SWV of bare SPCE and SPCE modified with ZIF-67 and ZIF-67/AgNWs were measured at different pH values. The bare or modified SPCE was immersed in a buffer containing 1 mM [Fe (CN)_6_]^3−/4−^ as a redox probe. The SWV curves of the ZIF-67/AgNWs@SPCE platform were measured in PBS containing FA under the optimum conditions. The final *C*_FA_ was regulated between 0.1 and 10 μM. The relationship between *I*_FA_ (redox peak current intensity of FA) and *C*_FA_ was plotted for use with the electrochemical sensor for accurate FA determination.

### 2.6. Selectivity, Stability, and Practicality of the Electrochemical Sensor

Under the optimum experimental conditions, the ZIF-67/AgNWs@SPCE sensing platform was used as an electrochemical sensor for determination of FA (1 μM). The signal responses obtained with potential interfering substances, including UA, AA, K^+^, Na^+^, Cl^−^, lactate, vitamin B_1_, vitamin B_2_ and glucose, were evaluated at a concentration of 0.1 mM. The value of *I*_FA_ was used to evaluate the selectivity of the sensor. The SWV signal was measured several times in succession in a buffer containing 1 μM FA. The stability was measured by the SWV method for 10 μM FA for 20 consecutive measurements. The sensor was also kept at 4 °C and tested every 24 h in the presence of FA (10 μM).

The practicality of the sensor was evaluated for determination of FA in real biological samples. Serum samples were diluted 10-fold with PBS (1 mM, pH = 7.4), and then a series of solutions was prepared with different concentrations of FA. The ZIF-67/AgNWs@SPCE sensing platform was immersed in each solution and the SWV curve was measured. The *C*_FA_ in each sample was calculated from the linear relationship between *I*_FA_ and *C*_FA_. Each measurement was repeated six times and the results are expressed as the mean ± standard deviation (SD).

## 3. Results and Discussion

### 3.1. Microscopic Morphological Characterization

The microstructural and morphological characteristics of the prepared nanomaterials were characterized by SEM. The ZIF-67 had a typical regular octahedron structure with a size of several hundred nanometers (Figure 1a). The ZIF-67/AgNWs composites had a pendant-like heterogeneous structure with the ZIF-67 nanoparticles located around the AgNWs (Figure 1b) [35]. Enlargement of the SEM images showed that the outer surface of ZIF-67/AgNWs was rough, whereas the surface of the blank ZIF-67 was relatively smooth. The binding of AgNWs to ZIF-67 did not affect the framework structure of ZIF-67. The AgNWs were homogeneous with a width of approximately 60 nm and lengths of 2–3 μm (Figure 1c). Furthermore, EDS was carried out showing the elemental composition of ZIF-67/AgNWs (Appendix A), which verified the presence of Co and Ag. Additionally, the weight and atomic percentages were described at the table of elemental compositions of ZIF-67/AgNWs in Appendix A inset. The characterization results indicated that the ZIF-67/AgNWs nanocomposite was successfully prepared.

### 3.2. Spectral Characterizations

The prepared nanomaterials were characterized using various spectroscopic techniques. UV-vis absorption spectra of ZIF-67 and the ZIF-67/AgNWs dispersed in methanol were obtained (Figure 2a) [36]. For ZIF-67 and the ZIF-67/AgNWs, the products showed characteristic absorption bands at 600 nm with Co^2+^ tetrahedral connections. The Raman spectrum of the ZIF-67/AgNWs was similar to that of pure ZIF-67 (Figure 2b). Several low-energy Raman bands (<500 cm^−1^) were derived from the vibrational modes of the metallic ligands, which indicated that the imidazole ligands coordinated with the Co center. Raman peaks were assigned to C-N vibrations (1143 cm^−1^), C-H vibrations (3132 cm^−1^, 1450 cm^−1^), and symmetric and asymmetric methyl C-H vibrations (2921 cm^−1^) [37]. The ZIF-67/AgNWs exhibited strong and broad absorption bands because of the synergistic effect arising from the modification of ZIF-67 with AgNWs, which greatly amplified the Surface-Enhanced Raman Scattering (SERS) signal [38]. In the FT-IR spectra for the two products (Figure 2c), bands at 690 cm^−1^ and 748 cm^−1^ were assigned to out-of-plane bending of the imidazole ring, and peaks at 990 cm^−1^ and 1303 cm^−1^ were assigned to in-plane bending. A peak at 1577 cm^−1^ was attributed to stretching of the imidazole ring, and a peak at 1697 cm^−1^ was assigned to N-H bending vibration. A band at 1300–1500 cm^−1^ was attributed to whole ring stretching. Peaks at 2924 cm^−1^ and 3136 cm^−1^ were assigned to aliphatic and aromatic C-H stretching of imidazole, respectively [35].

The XRD diffraction pattern of ZIF-67 (Figure 2d) matched the previously reported XRD pattern for ZIF-67 crystals. The XRD pattern of the ZIF-67/AgNWs was almost identical to that of ZIF-67 but with the addition of a characteristic peak for silver (JCPDS NO. 040783). These results indicated that the AgNWs were successfully combined with ZIF-67 and had a negligible effect on the crystal structure of ZIF-67 [35,38]. The XPS spectra of the ZIF-67/AgNWs showed characteristic peaks for Co 2p, O 1s, N 1s, C 1s, and Ag 3d (Figure 2e) [35,38]. For ZIF-67, no Ag 3d peak was present in the XPS spectrum. These results showed that the AgNWs were combined with ZIF-67. The XPS spectra of Co 2p showed fine binding energy features (Figure 2f). Bands at 796.28 and 780.78 eV were assigned to Co 2p^1/2^ and Co 2p^3/2^, respectively. These peaks showed that both Co^2+^ and Co^+^ valence states coexisted [35].

### 3.3. Electrochemical Performance of the Modified Electrode

The preparation procedure of ZIF-67/AgNWs@SPCE sensing platform was evaluated by measuring CV and EIS curves for SPCE with different modifications in buffers containing [Fe (CN)_6_]^3−/4−^ as redox probes. The peak current of ZIF-67@SPCE was lower than that of bare SPCE (Figure 3a), and this was attributed to the low conductivity of ZIF-67. The AgNWs@SPCE showed a slightly higher peak current than bare SPCE because of the good conductivity of the AgNWs. Compared with blank ZIF-67, the ZIF-67/AgNWs had a higher peak current. These results indicated there was a synergistic effect between ZIF-67 and the AgNWs promoted the electrical conductivity of the ZIF-67/AgNWs nanocomposite. The AgNWs increased the effective electrode area and conductivity of ZIF-67, which improved the charge transfer rate and provided rapid and sensitive electrochemical measurement. For the electrochemical determination of FA, the aniline structure of FA was oxidized to p-benzoquinone, generating an electron transfer and a characteristic oxidation peak [39]. ZIF-67/AgNWs@SPCE had the largest oxidation current compared to other modified electrodes (Appendix A). In addition, the single oxidation peak indicated that the process of FA oxidation was irreversible.

The interfacial properties of the different modified SPCE surfaces were assessed using EIS curves (Figure 3b). The charge transfer resistance (Rct) was calculated from the semicircle diameter of the Nyquist plot, and its value was influenced by the electron transfer kinetics of the redox probe at the electrode interface (Appendix A). Large semicircular regions appeared when the SPCE surface was modified with different materials, which indicated resistance to charge transfer. The ZIF-67@SPCE had the largest diameter and the AgNWs@SPCE had a slightly smaller diameter than bare SPCE. The ZIF-67/AgNWs@SPCE had the smallest diameter. The trends for the redox currents and Rct were consistent with the CV results, which further proved that the ZIF-67/AgNWs@SPCE sensing platform was successfully prepared.

### 3.4. Sensitivity of the Electrochemical Sensor

Under the optimum electrochemical measurement conditions at pH 7.4 (Appendix A), the SWV curves were measured using the ZIF-67/AgNWs@SPCE sensing platform in electrolyte solution with different concentrations of FA. The *I*_FA_, which peaked at 0.65 V, increased with gradual increases in the *C*_FA_ from 0.1 μM to 10 μM (Figure 4a,b). There was a good linear relationship (*I*_FA_ = −2.035 *C*_FA_ −0.5856, R^2^ = 0.993) between *I*_FA_ and *C*_FA_ (Figure 4c). The LOD calculated using a signal-to-noise ratio of three was approximately 30 nM. These results showed that the electrochemical sensor was highly sensitive for the determination of FA.

### 3.5. Selectivity of the Electrochemical Sensor

To investigate the selectivity of the electrochemical signal response of the FA electrochemical sensor, the SWV curves of ZIF-67/AgNWs@SPCE were measured with the target FA and potential interfering substances (UA, AA, K^+^, Na^+^, Cl^−^, lactate, vitamin B_1_, vitamin B_2_ and glucose). The selectivity and sensitivity of the electrochemical sensor for FA were assessed by comparing the *I*_FA_ values (Figure 5). The *I*_FA_ was slightly affected when each interfering substance (0.1 mM) was added, but 1 μM FA produced the largest changes in the *I*_FA_. The interfering substances showed a little effect on the *I*_FA_, even when they were presented at a concentration 100 times higher than FA. These results indicated that the sensor had high selectivity for FA determination and can effectively overcome the influence of potential interfering substances.

### 3.6. Stability of the Electrochemical Sensor

Twenty consecutive measurements of 10 μM FA were performed using the SWV method to test the stability of the sensor. The SWV signal remained stable during this process (Figure 6a). The sensor was also tested every 24 h in the presence of FA (10 μM), and the signal peak remained stable (Figure 6b). These results showed that the sensor maintained good stability over 7 days and multiple sensing cycles.

### 3.7. Determination Performance of the Electrochemical Sensor in Real Samples

The practicality of the electrochemical sensor was evaluated using real biological samples. The levels of FA in real serum samples with or without FA were determined from the *I*_FA_ and *C*_FA_ plot constructed for reference samples (Figure 4c). The *C*_FA_ assay values (Table 1) were close to the spiked values with high determination recovery rates (100.9–102.1%) and low RSDs (2.459–3.065%) (*n* = 6). The *C*_FA_ assay values agreed with the spiked values. These results indicated that the developed electrochemical sensor using the ZIF-67/AgNWs@SPCE sensing platform was capable of accurately capturing and detecting FA in real biofluid samples with high reliability and practicality.

### 3.8. Comparison of the Electrochemical Sensor with Other Methods

Compared with previous electrochemical methods (Table 2), the electrochemical sensing platform in this work demonstrated significant advantages in low concentration. The determination limit of the present sensor is 10–100 times less than most other platforms, and this could be attributed to the excellent electrochemical properties of the ZIF-67/AgNWs nanomaterials. Although the determination limits are similar to some other sensing platforms, these platforms are more complicated and expensive than our sensor. Moreover, our sensor had an excellent stability after a 7-day storage, which was benefited from the stability of ZIFs.

## 4. Conclusions

An electrochemical sensor was designed for highly sensitive and stable determination of FA. There was an excellent linear relationship between *I*_FA_ and *C*_FA_. The linear range for determination of FA was 0.1 μM to 10 μM and the determination limit was 30 nM. This electrochemical sensor showed high sensitivity for FA and maintained good stability over 7 days. Responses of the sensor to potential interfering substances were very weak. The sensor could also detect FA in human serum with a low RSD and high recovery. Our experimental results show that the electrochemical sensor is reliable and practical for the determination of FA in real samples.

## Data Availability

Data available on request from the authors.

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
