# Peer review of "Sensitive and Stable Electrochemical Sensor for Folic Acid Determination Using a ZIF-67/AgNWs Nanocomposite"

_biosensors, 2022, doi:10.3390/bios12060382_

Round 1
Reviewer 1 Report
In this work, an electrochemical sensor using silver nanowires-doped with a zeolite-like metal-organic framework (ZIF-67) was developed for the highly sensitive and stable determination of folic acid. The optimization of the analytical method for the determination of folic acid contains flaws and has not been fully validated. It is visible that the calibration plot does not have the necessary linearity that a calibration plot requires. Therefore, I recommend that the article be rejected for publication in Biosensors.
General comments:
1. A comparative study of the behavior of folic acid in bare SPCE and in modified SPCE was not presented. The analyte of this work is folic acid, so performing this comparative study using the [Fe(CN)6]3−/4− pair does not justify the modification of the electrode.
2. A modified electrode preparation optimization study (amount of silver nanowires and ZIF-67) was not presented.
3. Voltammograms must be presented in accordance with the IUPAC convention. It is very confusing to see an oxidation current with a negative sign.
4. The peak separation of the [Fe(CN)6]3−/4− pair is quite large. The electrodes used made the electrochemical reaction difficult. Also, voltammograms (Figure 3A) do not end up at the same potential value.
5. The Nyquist plot must be square in the scale and in the size, in order to observe the presence or absence of the double-layer capacitance or the presence of a constant phase element. Furthermore, no improvement in the value of Rct is seen by combining the silver nanowires with ZIF-67. There was no synergistic effect described by the authors. Nyquist chart axes are wrong. The correct one is –Z” vs. Z’.
6. The pH study was not discussed. Why was the best pH value 7.5? There is no discussion about it.
7. Why was SWV used? What were the SWV parameters used? Why were the parameters (frequency, increment, amplitude) not optimized? None of this was presented. There was no optimization of the electrochemical method.
8. It is visible that the calibration plot does not have the necessary linearity that a calibration plot requires. The value of the coefficient of determination shown cannot be right. It is visible that there is no linearity.
Reviewer 2 Report
The submitted manuscript describes an electrochemical sensor using silver nanowires (AgNWs)-doped with a zeolite-like metal-organic frameworks (ZIF-67) was developed for highly sensitive and stable detection of folic acid (FA). The ZIF- 67/AgNWs nanocomposite was prepared by a one-step reaction via a template method and drop-coated onto the surface of a screen-printed carbon electrode (SPCE) to form a ZIF-67/AgNWs@SPCE electrochemicalcal sensing platform. The electrochemical square wave voltammetry (SWV) curve for this sensing platform was measured in an electrolyte solution containing FA under the optimum experimental conditions. The redox peak current of FA (IFA) increased with increases in the FA concentration (CFA). There was a linear relationship between IFA and CFA in the range of 0.01 μM to 10 μM, and the detection limit was 30 nM. The ZIF-67/AgNWs@SPCE was used as an electrochemical sensor for FA which maintained good stability over 7 days and showed good detection performance in real samples with a high recovery rate (97.07%–99.64%, n = 6). Thus, I strongly recommend this manuscript for publication after major revision
Comments:
Introduction:
- Insert a new paragraph to explain the advantages of the used technique concerning other utilizing techniques.
- Zeolite imidazole ester skeletons (ZIFs) should be changed to Zeolite imidazole framework (ZIFs)
- Clarify the benefits of (ZIFs) and make it the desired choice over others
- You don’t mention some previous literature that detected folic acid using an electrochemical technique
- provide short notes about the difference between the studied mechanism with respect to other colorimetric or optical sensing mechanism mechanisms
Materials and methodology
- I suggest addressing the website of each supporting company (the links should be up to date) for both materials and instruments
Results and discussion
- The advantages of ZIFs in comparison with other methods should be highlighted, including analytical characteristics, reproducibility, specificity, stability
- Provide more details about the utilized mechanism?
- Explain the specified range of the chosen pH values
6.8-7.0-7.4-8.0
- 4- The interfering substances are limited can you choose other potential substances?
- I suggest measuring the FA using each potential interfering substance separately
Reviewer 3 Report
The report on the “Sensitive and Stable Electrochemical Sensor for Folic acid Detection using a ZIF-67/AgNWs Nanocomposite” The content of the work is interesting for readers from multidisciplinary disciplines including electrochemical sensors and biosensors. Still, the manuscript cannot be published in the present form due to the following issues:
- The introduction should be enhanced
- The energy bandgap is required to understand the effect of Ag NWs over ZIF-67
- EDAX is mandatory
- The equivalent circuit diagram is required in Figure 3 (b), Also the table of different entities is required such as Rct, etc.
- Grammar and many typological errors are present in the present form of the manuscript.
- The author should involve the novelty of the work which should be added before the conclusion part.
Round 2
Reviewer 1 Report
In this work, an electrochemical sensor using silver nanowires-doped with a zeolite-like metal-organic framework (ZIF-67) was developed for the highly sensitive and stable determination of folic acid. After the first review, the authors presented a new version of the manuscript. Some doubts were clarified and the manuscript was improved. However, there are still some points to be reviewed. Therefore, I recommend that a minor revision be carried out prior to publication in Biosensors.
General comments:
1. The authors, like many others, confuse the terms "detection" and "determination". Detection is qualitative by nature, while determination always is quantitative. Qualitative analysis is the detection of the presence of ions or compounds in an unknown sample, for example. The term "determination" refers to quantitative analysis to obtain data on the amount of analyte by weight or by the concentration of an element or a compound in a sample. Thus, the term "detection" in the manuscript (including in the title) should be replaced by "determination".
2. (Line 57). Folic acid is known as vitamin B9. Thus, one should describe what vitamins are. Add the sentence in the introduction: “Vitamins are organic substances found in foods and they are essential in small quantities for normal body functioning”. Add the Microchemical Journal, 179 (2022) 107588 reference to validate this information.
3. (Line 107). Do not use the acronym PBS for "phosphate-buffered solution". PBS is the accepted acronym for "phosphate-buffered saline" which contains 0.9% NaCl to warrant physiological ionic strength. See the Sigma Aldrich catalog (product P5368), for example. Also, see: https://en.wikipedia.org/wiki/Saline (medicine). Unfortunately, PBS is often wrongly used as an acronym for "phosphate buffer(ed) solution" in the literature but this is wrong and can cause confusion. Does the buffer employed by the authors really contain 0.9% NaCl (or other electrolytes such as MgCl2)? If yes, please specify. See: https://en.wikipedia.org/wiki/Phosphate-buffered_saline.
4. (Line 184). Check the grammar.
5. (Line 200). Check the −1.
6. (Lines 229-230). The sentence must be valid with a reference. Add a reference.
7. (Line 310). Reproducibility is the value below which the absolute difference between two single test results obtained by the same method on investigated test compounds under different conditions such as different analysts, different equipment, and different laboratories. For more information, see The Open Analytical Chemistry Journal, 5 (2011) 1–21. The authors performed repeatability tests.
8. Table 2 needs to be completed with recent works dedicated to the determination of folic acid. Please add the following works to the table: Journal of Food Composition and Analysis, 92 (2020) 103511 and Diamond and Related Materials, 121 (2022) 108713.
9. (Lines 311-312). Are the materials used for electrode preparation compatible with human skin? Have tests been carried out for this? If not, the claim of use as a wearable sensor should be excluded.
Reviewer 2 Report
The authors address all the required changes based on the reviewers' comments. Hereby, I recommend this manuscript for the publication in Biosensors Journal
Author Response
Thank you very much!
Reviewer 3 Report
The revised manuscript is acceptable for publication.
Author Response
Thank you very much!